# Everyday Memory Questionnaire—Revised (EMQ-R): Psychometric Validation of the European Portuguese Version in Non-Clinical Sample

**DOI:** 10.3390/bs15030280

**Published:** 2025-02-27

**Authors:** Pedro F. S. Rodrigues, Ana Bártolo, Bruna Ribeiro, Ramón López-Higes, Susana Rubio-Valdehita, Ana Paula Caetano, Sara M. Fernandes

**Affiliations:** 1CINTESIS.UPT@RISE-Health, Portucalense University, 4200-072 Porto, Portugal; 2Department of Psychology and Education, Portucalense University, 4200-072 Porto, Portugal; 3Departamento de Psicología Experimental, Complutense University of Madrid (UCM), 28223 Madrid, Spain; 4Departamento de Psicología Social, del Trabajo y Diferencial, Complutense University of Madrid (UCM), 28223 Madrid, Spain; 5Intrepid Lab, Lusófona University, 4000-098 Porto, Portugal; 6CLISSIS, Lusiada Research Center on Social Work and Social Intervention, 1349-001 Lisboa, Portugal

**Keywords:** EMQ-R, memory complaints, non-clinical, screening tool, psychometric validation

## Abstract

The present study aimed to translate, culturally adapt, and present a psychometric validation for the Everyday Memory Questionnaire—Revised (EMQ-R) to the Portuguese population. The study involved 267 participants aged between 18 and 75 years (*M* = 39.32; *SD* = 14.8), recruited online. Self-report measures of anxiety and depression symptoms were administered to assess the instrument’s convergent validity. To examine the factorial structure of the measure, a two-step validation process was employed. Given the uncertainty about the optimal measurement model, the sample was randomly divided into two independent subsamples. First, a principal component analysis (PCA) was conducted to explore the factorial structure. Next, a confirmatory factor analysis (CFA) was performed to validate the identified structure. The results supported a unidimensional structure consisting of 12 items, suggesting that perceived memory difficulties are best represented as a single overarching factor. High reliability was observed for this structure (Cronbach’s alpha and McDonald’s omega values ≥ 0.90). The results also indicated that general memory complaints were moderately correlated with symptoms of anxiety and depression. Furthermore, the study highlighted the promising potential of the measure as a screening tool for detecting subjective memory complaints, with an optimal cut-off score of 16 points. Future studies should focus on validating the EMQ-R with clinical samples, exploring its discriminative ability, and examining the stability of the cut-off score across different populations and contexts.

## 1. Introduction

Memory is a fundamental cognitive function that enables individuals to encode, store, and retrieve information, serving as a basis for daily functioning and overall quality of life ([2]). When disruptions in memory processes occur, they can substantially impair an individual’s ability to perform everyday tasks, maintain social relationships, and achieve personal goals ([37]). Subjective memory complaints (SMCs) refer to self-reported difficulties or concerns about memory performance that may or may not align with objective memory impairments. The Everyday Memory Questionnaire—Revised (EMQ-R) is closely linked to the evaluation of SMCs, as it allows individuals to reflect on their daily memory challenges in a variety of real-world contexts, providing a structured framework to quantify these perceptions. The accurate assessment of everyday memory complaints has become a crucial focus in both clinical and research settings. Among the instruments developed for this purpose, the Everyday Memory Questionnaire—Revised (EMQ-R) stands out as a widely used self-report tool designed to evaluate memory lapses in real-world contexts ([22]). Despite its popularity and robust psychometric properties in its original English version, the EMQ-R’s applicability across diverse linguistic and cultural contexts remains uncertain, necessitating thorough validation efforts.

The EMQ-R is a revision of the original Everyday Memory Questionnaire (EMQ), first introduced by [29] ([29]). The revised version simplifies the original instrument by reducing the number of items (13 items) and enhancing its psychometric properties while retaining its core purpose of assessing everyday memory failures ([22]). This self-report instrument (with two main factors: retrieval and attentional tracking) asks individuals to reflect on the frequency of memory failures in various real-world scenarios, such as forgetting names, appointments, or tasks. The revised version improves the psychometric properties of the original and ensures better applicability across different populations. The EMQ-R is particularly useful in clinical settings to identify functional memory difficulties in diverse populations, including individuals with neurological conditions, individuals with psychiatric disorders, and healthy controls ([38]; [22]). Indeed, the EMQ-R has been validated in several countries (e.g., [1]; [28]; [30]) and in different contexts (e.g., [23]; [38]). However, as with any psychometric tool, the validity and reliability of the EMQ-R cannot be assumed to generalize across languages and cultures without empirical validation. Translation and cultural adaptation are essential to ensure that the instrument measures the intended constructs consistently and accurately in new contexts ([40]).

Specifically, Portugal has aging populations that are increasingly affected by conditions such as Alzheimer’s disease and other forms of dementia ([34]). Everyday memory complaints are among the earliest and most common symptoms reported by individuals with cognitive decline, making tools like the EMQ-R invaluable for early detection and intervention ([33]). Furthermore, everyday memory assessment can provide insights into the functional impacts of memory impairments, complementing neuropsychological tests that focus on laboratory-based measures of cognitive performance ([8]). Interestingly, memory complaints are present in several age groups, although their nature may differ qualitatively (for instance in young and older adults; [11]). Therefore, it is essential to validate and apply instruments measuring memory failures within specific populations to ensure reliable and accurate assessments.

The process of psychometric validation involves multiple steps, including translation, cultural adaptation, and the evaluation of the instrument’s reliability and validity in the target population ([26]). Reliability refers to the consistency of the instrument’s measurements, often assessed through internal consistency and test-retest reliability ([13]). Validity, on the other hand, pertains to the degree to which the instrument measures what it purports to measure and includes aspects such as content validity, construct validity, and criterion-related validity ([9]). In the case of the Portuguese version of the EMQ-R, it is critical to comprehensively evaluate these properties to ensure that the tool is both reliable and valid for Portuguese-speaking populations.

Previous studies on the adaptation of psychometric instruments into European Portuguese have highlighted several challenges and considerations. The importance of validating the Portuguese version of the EMQ-R extends beyond clinical and research applications. Everyday memory is a multidimensional construct that intersects with various domains of functioning, including emotional well-being, occupational performance, and social relationships ([31]). Indeed, the literature has been relatively consistent in stating that symptoms of depression and anxiety are strongly associated with subjective memory complaints ([5]; [25]; [35]; [36]), although the specific relationship (e.g., cause–effect) is unknown (e.g., [3]; [36]). Understanding the patterns and predictors of everyday memory failures can inform interventions aimed at improving cognitive health and overall quality of life. However, instruments with adequate psychometric properties are necessary to assess memory complaints and, consequently, the relationships between these and other psychological factors. The availability of a validated Portuguese version of the EMQ-R would facilitate cross-cultural research on memory and cognition, enabling comparisons between Portuguese-speaking populations and other groups worldwide. Such research is essential for identifying universal and culture-specific factors that influence everyday memory.

Despite the clear need for psychometric validation, few studies have focused on the adaptation of everyday memory assessment tools for the Portuguese population. Some exceptions include the validation of instruments like the Prospective and Retrospective Memory Questionnaire (PRMQ) and the Subjective Memory Complaints ([10]; [19]). Specifically, instruments that assess subjective memory complaints in Portugal have shown very poor psychometric properties. These studies have highlighted both the feasibility and the challenges of adapting memory assessment tools, underscoring the importance of rigorous methodology and comprehensive evaluation. Building on this foundation, the present study aims to address a significant gap in the literature by validating the European Portuguese version of the EMQ-R in a non-clinical sample.

Based on these considerations, the current study aimed to investigate the following: (i) translate and culturally adapt the Everyday Memory Questionnaire—Revised (EMQ-R) into European Portuguese; (ii) explore the reliability and construct validity of the measure (e.g., factor validity and convergent validity); and (iii) evaluate the screening performance of the EMQ-R in identifying individuals with perceived memory difficulties and determine the optimal cut-off scores.

## 2. Materials and Methods

### 2.1. Participants and Procedure

To determine the minimum sample size required to assess the structural validity of the EMQ-R, the COSMIN guidelines ([17]) were followed, which recommend a ratio of seven participants per variable. Since the initial EMQ-R scale consists of 13 items, a minimum sample size of 91 participants (7 × 13 items = 91) would be required. The sample was non-probabilistic, and individuals were considered eligible to participate if they self-declared: (a) Portuguese-speaking; (b) aged between 18 and 75 years; and (c) no history of objective cognitive deficits and/or brain injury. Participant recruitment was conducted online through the distribution of the questionnaire via an online platform. The study was advertised on pages and groups across various social media networks (e.g., Facebook and Instagram), as well as through emails sent to higher education institutions. Prior to participating in the study, all participants were provided with an informed consent form, which they were required to accept by selecting a checkbox. All ethical research procedures were followed according to the Code of Ethics of Portuguese Psychologists and the Helsinki Declaration, ensuring the anonymity and confidentiality of the participants.

### 2.2. Measures

The assessment protocol included a sociodemographic and clinical questionnaire designed to collect personal information such as sex, age, marital status, education, psychiatric history, and perceived memory difficulties. The following self-report measures were administered:

Everyday Memory Questionnaire—Revised (EMQ-R; [22]): The EMQ-R is a shortened version of the Everyday Memory Questionnaire, which was initially developed for use with survivors of head injuries ([29]) and later applied to both non-clinical and clinical samples. The original shortened version consists of 13 items (e.g., “Forgetting when it was that something happened; for example, whether it was yesterday or last week”), rated on a Likert-type scale ranging from “once or less in the last month” (0 points) to “once or more once per day” (4 points). The 13 items are organized into 3 factors: (F1) Retrieval, (F2) Attentional Tracking, and (F3) an unspecified factor. The EMQ-R has demonstrated strong psychometric properties in assessing individuals’ perceptions of memory performance in everyday life, even in healthy individuals. Following international and consensual guidelines for translation and validation of psychological instruments (e.g., [39], [40]), the original version of the EMQ-R was translated into European Portuguese. The translation was carried out by two bilingual translators independently, both fluent in Portuguese and English. This ensured knowledge of both languages, cultural nuances, and the terminology of the scale’s constructs to guarantee that the items maintained the same meaning as the original version. Based on these two versions, a first consolidated version was created, which was subsequently backtranslated. This consolidated version of the measure was reviewed by a panel of experts in the fields of Cognitive Psychology and Neuropsychology (*n* = 5). The experts had the opportunity to conduct an independent review, with suggestions for semantic and cultural adjustments. All proposed suggestions were incorporated, and a pre-test was conducted with a community sample (*n* = 10). The European Portuguese version of the EMQ-R is available in Appendix A.

Beck Depression Inventory-II (BDI-II; [4]; [7]): The BDI-II contains 21 items, with a scale ranging from zero to three points, zero being the absence of symptoms and three being the presence of strong symptomatology. For each statement, participants were instructed to choose the option that best fits their situation during the previous two weeks, including the day they completed the questionnaire. The BDI-II is a reliable index for assessing the severity of depressive symptomatology, with robust psychometric properties for the Portuguese population, including an internal consistency Cronbach alpha of 0.91. This instrument includes a three-factor structure: cognitive, affective, and somatic factors ([18]).

State-Trait Anxiety Inventory (STAI-Y2; [27]; [24]): The STAI-Y2 was developed to measure an individual’s tendency to experience anxiety. The instrument consists of 40 items, divided into two subscales: 20 that compose the “state” subscale (Y1) and 20 that correspond to the “trait” subscale (Y2). The Y1 subscale measures a transitory emotion characterized by physiological arousal and feelings of apprehension, dread, and tension. In contrast, the Y2 assesses a stable tendency to perceive situations as threatening or stressful. In the present study, the Y2 subscale was applied, using a Likert-type scale ranging from one to four points, with higher scores indicating higher levels of anxiety. The European Portuguese version of the scale has shown good internal consistency, with a Cronbach alpha ≥ 0.87 ([24]).

### 2.3. Data Analysis

Statistical analysis was conducted using the Statistical Package for Social Sciences, version 29 (SPSS Inc., Chicago), and MPlus, version 6.12 (Muthén & Muthén, Los Angeles, CA, USA). Descriptive analyses were performed to characterize the study sample and the properties of the EMQ-R items (e.g., medians, interquartile range, and normality assumptions). The psychometric properties of the measure were examined in terms of factorial validity, reliability, and convergent validity. The factor structure of the EMQ-R was examined using a two-step validation process. Given the uncertainty regarding the most appropriate measurement model for this scale, the sample was randomly divided into two independent subsamples. This approach ensured a more rigorous assessment of the factorial validity by combining exploratory and confirmatory techniques. A component principal analysis (PCA) was conducted on the first subsample. A principal component analysis (PCA) was conducted on the first subsample. Then, the factor structure identified in the EFA was tested in the second subsample using confirmatory factor analysis (CFA). Model fit was assessed through multiple indices. The chi-square test (*χ*^2^) was used to assess the magnitude of the discrepancy between the hypothesized and observed models. A non-significant *χ*^2^ is indicative of good model fit; however, due to the sensitivity of *χ*^2^ to large sample sizes ([32]), additional fit indices were also evaluated, including: (i) the comparative fit index (CFI), (ii) root mean square error of approximation (RMSEA), and (iii) weighted root mean square residual (WRMR), generated by MPlus. A model was considered to have an acceptable fit if *χ*^2^*/df* ≤ 5, RMSEA ≤ 0.10, CFI ≥ 0.90, and WRMR < 1.00 ([12]; [15]).

Reliability was assessed using Cronbach’s alpha and McDonald’s omega to ensure robustness in the analysis. Convergent validity was examined through bivariate correlations between EMQ-R scores and measures of anxiety (STAI) and depression (BDI), constructs theoretically associated with subjective memory complaints. Pearson’s correlation coefficient was used, and the strength of correlations was classified as weak (0–0.3), moderate (0.3–0.7), or strong (0.7–1.0), following [20] ([20]) guidelines.

To evaluate the screening performance of the EMQ-R in identifying individuals with subjective memory complaints and to determine optimal cut-off scores, receiver operating characteristic (ROC) curve analysis was conducted. This analysis utilized a binary criterion derived from a single stated item in which participants were asked whether they perceived memory difficulties in their daily lives (e.g., see Table 1). The area under the ROC curve (AUC) was calculated to quantify the screening ability of the EMQ-R and was compared with the diagonal line (AUC = 0.50), which represents classification by chance. The optimal cut-off score was identified as the closest point to the intersection of the ROC curve with the diagonal line extending from the upper left to the lower right side of the graph, balancing sensitivity and specificity equally. The Youden index (J) was also computed as a summary measure of the ROC curve to determine the screening ability and optimal threshold for the EMQ-R.

## 3. Results

### 3.1. Characterization of Study Participants

A total of 267 participants were included in this study. Table 1 summarizes the sociodemographic and clinical characteristics of the sample. The participants’ mean age was 39.32 years (*SD* = 14.8), ranging from 18 to 75 years. Most participants were women (72.7%; *n* = 194). In terms of education, most participants had completed high school (34.5%; *n* = 92) or university (28.5%; *n* = 76). Regarding marital status, 47.9% (*n* = 128) were single, while 35.2% (*n* = 94) were married. Among individuals, 35.6% (*n* = 95) reported experiencing current memory complaints.

### 3.2. Assessment of Item Properties

All possible response values on the Likert scale (0–4) were observed for each item. The median for most items was close to one (see Table 2), except for item 1 (*Mdn* = 2), related to checking if a task has been completed, and items 4 and 9 (Mdn = 0), related to unknowingly rereading something and losing track of a story while reading, respectively. No significant deviations from normality were found, considering the absolute values of skewness (*Sk*, <3.0) and kurtosis (Ku, <7.0; [6]). All items showed corrected item-total correlations above 0.30 (0.481 ≤ *r* ≤ 0.761), indicating that they adequately contribute to the instrument’s internal consistency. No substantial increase in Cronbach’s alpha was observed when any item was excluded. Inter-item correlations were mostly above 0.30, with no correlations exceeding 0.80, indicating that the items are sufficiently distinct without significant redundancy (0.235 ≤ r ≤ 0.703; see Appendix B). Item 4 showed the lowest correlations with other items.

### 3.3. Factor Validity (Part I): Principal Component Analysis (PCA)

To examine the factor structure, a PCA was conducted using a randomized split of the sample data (*n* = 133). Following the original validation procedure, an oblimin rotation was applied. The Kaiser–Meyer–Olkin (KMO) measure indicated high sampling adequacy (0.91), and Bartlett’s test of sphericity was significant (*χ*^2^(78) = 1155.18, *p* < 0.001), confirming the suitability of the correlation matrix. Using the Kaiser criterion (eigenvalues > 1.00), the analysis identified a single factor accounting for 57.44% of the total variance. This structure differs from the original, which suggested three factors ([22]). Table 3 presents the final factor pattern matrix and item communalities. All items had factor loadings above 0.50. Items 1 and 4 showed the lowest communalities but were retained in the measure’s structure. Considering the unifactorial solution, Cronbach’s alpha and McDonald’s Omega indicated good internal consistency.

### 3.4. Factor Validity (Part II): Confirmatory Factor Analysis (CFA)

The one-factor model derived from the PCA was then cross-validated on a subsample of 134 participants. Given the ordinal nature of the EMQ-R, a confirmatory factor analysis (CFA) was conducted using the weighted least squares mean and variance adjusted (WLSMV) estimator (e.g., [14]). Model fit was assessed using multiple indices. The *χ*^2^/*df* ratio was 2.68, indicating an acceptable level of discrepancy between the model and the data. The CFI was 0.95, suggesting good relative fit. The RMSEA was 0.11 (90% CI: 0.09–0.13), indicating a marginal fit with some degree of model misspecification. The WRMR was 0.96, falling within the recommended threshold for a good fit. While most indices suggest an adequate model fit, the elevated RMSEA value and potential model misspecifications warranted further examination. To refine the model, modification indices were analyzed, and re-specifications were considered to improve the overall fit. The analysis of modification indices led to the decision to exclude Item 4 [“Starting to read something (e.g., a book, a newspaper article, or a magazine) without realizing that it had already been read”]. The correlation between the errors of item 4 and other variables indicated that it might not be well represented by the latent factor being analyzed, potentially diluting the impact of the factor and compromising the theoretical coherence of the factor structure. Additionally, item 4 had the lowest factor loading among the items, suggesting a relatively smaller contribution to the latent factor, which justified its removal to improve the model’s precision. In the previous PCA, this item had already shown low communality, further supporting this decision. The exclusion of item 4 resulted in improved fit indices, as shown by the increase in CFI (from 0.948 to 0.959) and the decrease in WRMR (from 0.958 to 0.882; see Table 4). These improvements indicate that excluding this item contributed to a better model fit. The RMSEA also showed a slight improvement; although it remains above the ideal value, it is still within an acceptable range (0.09–0.10; Kline, 2005). Factor loadings for the model were all statistically significant, with estimates ranging from 0.63 to 0.79 (see Figure 1).

### 3.5. Convergent Validity

The total EMQ-R score showed moderate correlations with anxiety and depressive symptoms, suggesting that higher levels of these symptoms are associated with greater memory complaints. These findings support the convergent validity of the EMQ-R by demonstrating its association with theoretically related constructs (see Table 5).

### 3.6. Screening Performance and Cut-Off Score

Considering a 12-item structure for the Portuguese version, based on the best structure proposed by the factor analysis, the total scores for the Portuguese version of the EMQ-R can range from 0 to 48 points, with higher scores indicating greater general memory complaints. Figure 2 presents the ROC curve analysis conducted to assess the screening performance of the measure in identifying individuals with perceived memory difficulties and to determine the optimal cut-off score.

The AUC for the EMQ-R total score was 0.61 (95% CI: 0.54–0.68), which is significantly higher (*p* = 0.003) than the diagonal line, indicating that the EMQ-R performs better than random chance in distinguishing between individuals with and without memory complaints. The optimal cut-off score for the EMQ-R was identified at ≥15.5, which was rounded to 16. This cut-off was closest to the intersection point of the ROC curve and the diagonal line (see Figure 2). The Youden index confirmed that a cut-off of 16 yielded the highest index (*J* = 0.209). This score provides a balance between sensitivity (56%) and specificity (65%), both of which are considered acceptable.

## 4. Discussion

The present study aimed to translate and culturally adapt the Everyday Memory Questionnaire—Revised (EMQ-R) into European Portuguese and assess its psychometric properties using a community sample. Overall, our findings support the reliability and validity of the Portuguese version of the EMQ-R, highlighting its potential as a valuable tool for evaluating memory complaints.

Despite these promising results, differences were observed compared to the original study ([22]), particularly regarding the factorial structure. The confirmatory factor analysis (CFA) revealed that the original three-factor structure did not provide an adequate fit for the data in the Portuguese context. Instead, the results supported a unidimensional structure consisting of 12 items, suggesting that perceived memory difficulties are best represented as a single overarching factor, rather than distinct subcomponents.

Moreover, the original factorial structure had not been reassessed in previous studies to verify its adequacy, and it included a third factor with only two items, which lacked a clear conceptual interpretation ([22]). The present findings address this limitation by adopting a more integrated perspective, where memory complaints are understood as manifestations of a single underlying construct. This approach aligns with theoretical perspectives that emphasize the interconnected nature of memory processes ([2]), reinforcing the idea that perceived memory difficulties in daily life stem from overlapping cognitive mechanisms rather than independent deficits. Furthermore, the reliability analysis provided strong support for this unidimensional structure, demonstrating high internal consistency for the overall scale. These results reinforce the robustness and reliability of the Portuguese version of the EMQ-R in assessing perceived memory difficulties within this context.

In line with the literature, memory complaints, as measured by the scale, were found to be associated with symptoms of anxiety and depression. This finding has been consistently observed across different populations, life stages (e.g., [16]), and health conditions (e.g., [21]), further supporting the measure’s convergent validity.

Another key result of the study was related to assessing the screening performance of the EMQ-R. The results of the receiver operating characteristic (ROC) analysis suggested that the EMQ-R has a moderate ability to distinguish between individuals with and without perceived memory difficulties. The optimal cut-off score for the EMQ-R was identified as 16, based on the intersection of the ROC curve and the diagonal line. However, the sensitivity was 56%, and specificity was 65%, indicating that the EMQ-R is better at identifying individuals without memory complaints (specificity) than those with complaints (sensitivity). These findings suggest that, while the EMQ-R can be useful for screening, there is room for improvement, particularly in enhancing its sensitivity.

The promising results regarding the psychometric properties of the EMQ-R must be considered within the context of several study limitations. First, the sample used was non-probabilistic and consisted of individuals from the general community, which may limit the generalizability of the results to clinical populations or other cultural contexts. Future research should focus on validating the EMQ-R with clinical samples, particularly individuals with neurological conditions or diagnosed memory impairments, as in the original validation study by [22] ([22]), to assess the measure’s discriminative ability. Additionally, a more balanced distribution of age groups would be valuable for further exploring the measure’s discriminant validity. Future studies should also examine measurement invariance (MI) and differential item functioning (DIF) to determine whether the EMQ-R operates equivalently across different demographic and clinical groups. This would provide further evidence of the scale’s robustness and applicability across populations.

Second, although a cut-off score of 16 was identified in this study as an indicator of memory complaint differentiation, further studies are needed to evaluate the stability of this cut-off across different populations and settings. Investigating the potential to improve the measure’s sensitivity while maintaining specificity would also be beneficial. This could be achieved by using more reliable methods to assess perceived memory difficulties. Notably, a single stated item was used for the ROC curve analysis, which may limit the robustness of the results. The reliance on self-reported memory complaints as the true condition introduces a subjective bias, as self-reports can be influenced by individual perception, emotional state, and cognitive biases. Unlike clinical diagnoses, which provide objective measures of memory impairment, self-reported data cannot confirm actual cognitive deficits. Thus, future studies should incorporate clinical assessments to validate self-reported memory complaints and enhance the accuracy and reliability of the results.

It is also important to highlight that this study does not provide data on test-retest reliability. Finally, the convergent validity could be verified with other subjective memory measures, such as those used in other validation studies of the EMQ-R (e.g., [30]); however, in Portugal, the scarcity of such alternative instruments prevented us from pursuing this procedural option. Future research should address these gaps to ensure the stability and reliability of the EMQ-R over time.

## Figures and Tables

**Figure 1 behavsci-15-00280-f001:**
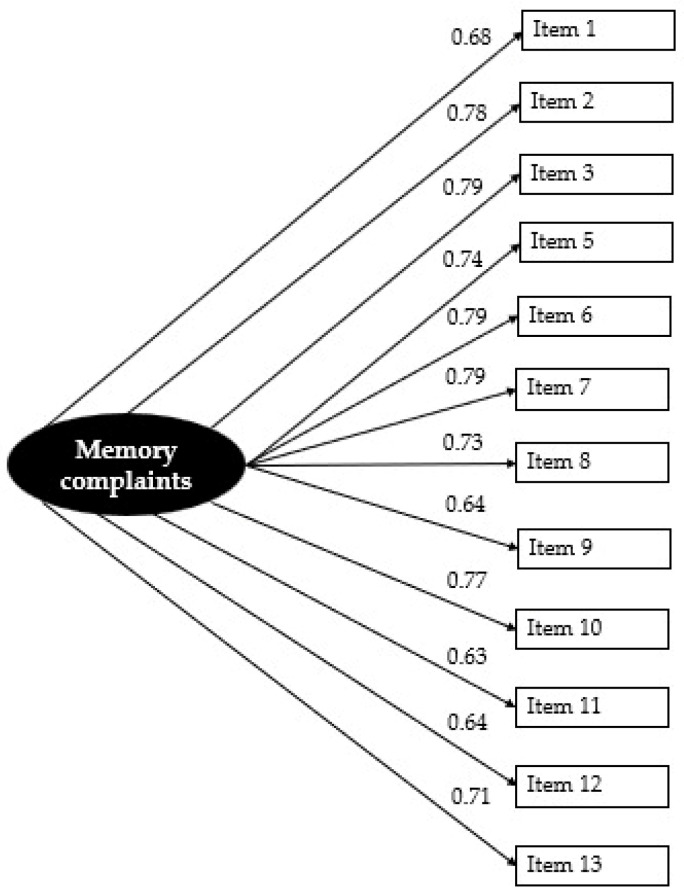
Standardized factor loadings for the one-factor model.

**Figure 2 behavsci-15-00280-f002:**
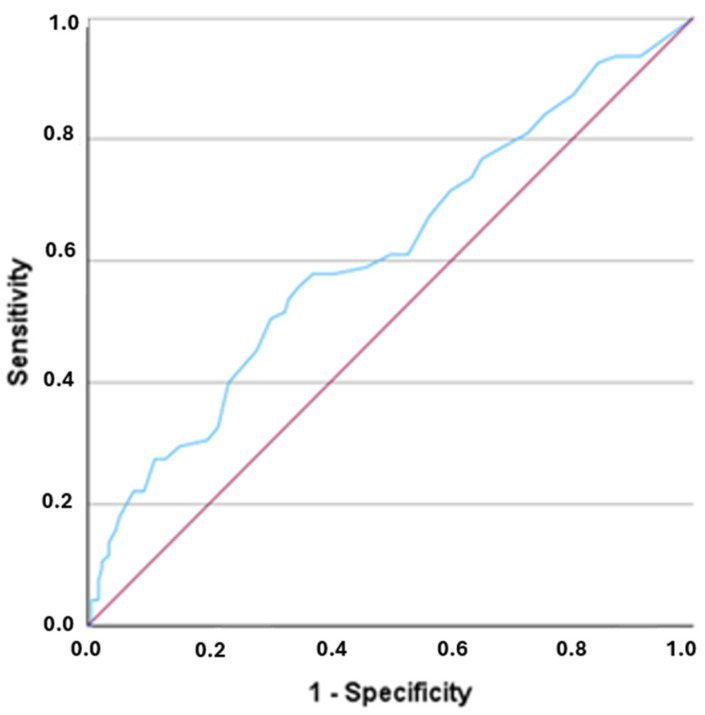
Receiver operating characteristic (ROC) curve of the EMQ-R.

**Table 1 behavsci-15-00280-t001:** Summary of sample characteristics (*N* = 267).

	*n*	%
Sex		
Male	73	27.3
Female	194	72.7
Age (M ± SD, range)	39.32 ± 14.8, 18–75	
Marital status		
Married	94	35.2
Cohabiting	22	8.2
Divorced/separated	14	5.2
Single	128	47.9
Widowed	9	3.4
Education		
Primary school	31	11.6
Middle school	68	25.5
High school	92	34.5
University	76	28.5
Current memory complaints (yes)	95	35.6

**Table 2 behavsci-15-00280-t002:** Preliminary analysis: descriptive statistics of EMQ-R items.

Item	Min–Max	Mdn (IQR)	M (SD)	Skewness	Kurtosis	Corrected Item-Total Correlation	Cronbach’s Alpha If Item Deleted
1	0–4	2 (2)	1.75 (1.40)	0.055	−1.272	0.559	0.921
2	0–4	1 (2)	1.28 (1.28)	0.433	−0.947	0.694	0.915
3	0–4	1 (2)	1.23 (1.28)	0.657	−0.610	0.740	0.913
4	0–4	0 (1)	0.71 (1.19)	1.477	1.008	0.481	0.922
5	0–4	1 (2)	1.44 (1.26)	0.449	−0.855	0.697	0.915
6	0–4	1 (2)	0.97 (1.10)	0.782	−0.435	0.735	0.914
7	0–4	1 (2)	0.93 (1.15)	0.877	−0.227	0.761	0.913
8	0–4	1 (2)	1.11 (1.24)	0.712	−0.722	0.697	0.915
9	0–4	0 (2)	0.91 (1.19)	1.106	0.063	0.650	0.917
10	0–4	1 (2)	1.02 (1.13)	0.838	−0.262	0.713	0.915
11	0–4	1 (2)	0.98 (1.11)	1.142	0.480	0.584	0.919
12	0–4	1 (3)	1.32 (1.33)	0.565	−0.937	0.659	0.917
13	0–4	1 (2)	0.92 (1.14)	1.004	0.083	0.659	0.916

Note: Mdn = Median; IQR = Interquartile range.

**Table 3 behavsci-15-00280-t003:** Principal component analysis: factor loadings, communalities, and internal consistency (*n* = 133).

Items	F1	h^2^
1	0.570	0.325
2	0.772	0.596
3	0.826	0.682
4	0.547	0.299
5	0.779	0.607
6	0.821	0.674
7	0.833	0.694
8	0.802	0.644
9	0.808	0.653
10	0.805	0.648
11	0.697	0.486
12	0.777	0.604
13	0.746	0.556
% Explained variance	57.44	
Cronbach’s alpha (α)	0.94	
McDonald’s omega (ω)	0.94	

**Table 4 behavsci-15-00280-t004:** Summary of the tested factor models: CFA analysis.

Model	*χ* ^2^	*χ*^2^/*df*	CFI	RMSEA (90% CI)	WRMR
▪ One-dimensional solution	174.228 ***	2.68	0.948	0.112 (0.092–0.132)	0.958
▪ One-dimensional solution, excluding item 4	139.672 ***	2.59	0.959	0.109 (0.087–0.131)	0.882

Note: *** *p* < 0.001; *χ*^2^ = Chi-square; *df* = degrees of freedom; CFI = Comparative Fit Index; RMSEA = Root Mean Square Error of Approximation; WRMR = Weighted Root Mean Square Residual.

**Table 5 behavsci-15-00280-t005:** Convergent validity: correlations between memory complaints and symptoms of anxiety and depression.

	M	SD	1.	2.	3.
1. EQM-R (total score)	14.42	10.70	1		
2. STAI	43.37	10.03	0.363 *	1	
3. BDI	14.06	11.34	0.400 **	0.691 ***	1

* *p* < 0.05; ** *p* < 0.01; *** *p* < 0.001.

## Data Availability

The raw data supporting the conclusions of this article will be made available by the authors on request due to privacy reasons.

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
