# Peer review of "Everyday Memory Questionnaire—Revised (EMQ-R): Psychometric Validation of the European Portuguese Version in Non-Clinical Sample"

_behavsci, 2025, doi:10.3390/bs15030280_

Round 1

Reviewer 1 Report

Comments and Suggestions for Authors

The current manuscript translated the everyday memory questionnaire (EMQ-R) to Portuguese and examined its psychometric properties. I think It is an important and practical issue that should be addressed. However, I have several concerns/suggestions about the manuscript. I listed them below for the authors as references.

1.      In the manuscript, the authors used BDI and STAI as tools to assess the convergent validity of the Portuguese version of EMQ-R. I wonder if is there any other research that tries to validate the EMQ-R using similar measures to assess the convergent validity of EMQ-R that can back up the authors’ decisions here.

As the author mentioned in the introduction, memory is an important cognitive ability. Thus. Is there any reason not to choose other measures that also focus on participants’ cognition functions as other studies (e.g., Taleb, Ismail, & Abou-Abbas, 2024)?

2.      Maybe I misunderstood some content of the manuscript, I wonder what is binary (criteria the authors used in the ROC curve analyses for the EMQ-R? Is it the diagnosis for each participant provided by clinical neuropsychologists? If so, I will suggest authors mention it in the manuscript.

3.      I also have several concerns about the factor analyses in the manuscript. It looks like the second-order CFA model also has some issues with the correlations between first-level factors. The standardized loadings from the higher factor to the two lower-level factors are both higher than 0.95, which means the correlation between them is still above 0.9. In other words, these two factors are not differentiable in CFA (i.e., similar to the multicollinearity issue the authors mentioned in the first paragraph of section 3.3).

Besides, the modification index will only tell researchers the estimated change in global Chisq after releasing one estimate constraint, rather than suggesting any kind of “second-order model”, if the starting model is a typical first-order CFA model.

Therefore, I will suggest the authors redo the analyses here and use more precise phrases to describe the procedures of FA. For example, given there is some uncertainty about the proper measurement model for the EMQ-R, I will suggest the authors split the sample into two groups. Using the first group to run EFA and explore the possibilities, and then use data from the second group to verify it with CFA.

In addition, the other can include indices that account for model complexity such as AIC and BIC in their report for CFA. These indices are available in Mplusin Mplus if the authors use ML estimators for categorical variables.

4.      To benefit the readers, I will suggest the authors directly provide the translated version of EMQ-R in the paper or at least in some online supplementary material (e.g., posted on OSF), rather than having readers further request for it via personal communications.

5.      Other minor issues, as the common practices of reporting FA results, I will suggest the authors add means and SDs of 13 items into Table 1. Besides, I will also recommend the authors provide the correlation matrix between these 13 items in the data, so other researchers can replicate the FA results to some degree.

6.      If all the participants are non-clinical persons, I suggest the authors add this information to the title of the article.

Reference

Taleb, A., Ismail, A., & Abou-Abbas, L. (2024). Psychometric properties of the Arabic version of the everyday memory questionnaire-revised (EMQ-R) among the Lebanese population. The Clinical Neuropsychologist, 1-18.

Author Response

Reviewer 1

The current manuscript translated the everyday memory questionnaire (EMQ-R) to Portuguese and examined its psychometric properties. I think It is an important and practical issue that should be addressed. However, I have several concerns/suggestions about the manuscript. I listed them below for the authors as references.

Our Answer (A): We sincerely appreciate the comments and valuable suggestions. Your insightful feedback has greatly contributed to improving and enriching our manuscript. Thank you for your time and effort in reviewing our work.

In the manuscript, the authors used BDI and STAI as tools to assess the convergent validity of the Portuguese version of EMQ-R. I wonder if is there any other research that tries to validate the EMQ-R using similar measures to assess the convergent validity of EMQ-R that can back up the authors’ decisions here.

A: Your comment is very important and highly relevant. Unfortunately, in Portugal, there are very few instruments available to assess subjective memory complaints; additionally, these present poor psychometric properties. Therefore, given the need to present psychometric characteristics of this kind of instruments, we decided to translate and conduct an exploratory validation of the EMQ-R, even though we do not have the direct possibility of establishing convergent validity. In other words, this serves as a starting point for future validation studies of similar instruments. The choice of depressive and anxiety-related symptomatology as a measure of convergent validity is based on the fact that numerous studies have consistently shown established relationships between subjective memory complaints and depression and anxiety (e.g. Brigola et al., 2015; Sousa et al., 2015; Yates et al., 2017, Zapater-Fajariet al., 2022). Now, we clarified this idea in the introduction and discussion (see Pages 2-3, Lines 98-116, text highlighted in yellow).

New References added in the manuscript:

Brigola, A. G., Manzini, C. S. S., Oliveira, G. B. S., Ottaviani, A. C., Sako, M. P., & Vale, F. A. C. (2015). Subjective memory complaints associated with depression and cognitive impairment in the elderly: A systematic review. Dementia & Neuropsychologia9(1), 51–57. https://doi.org/10.1590/S1980-57642015DN91000009

Sousa, M., Pereira, A., & Costa, R. (2015). Subjective Memory Complaint and Depressive Symptoms among Older Adults in Portugal. Current Gerontology and Geriatrics Research2015, 296581. https://doi.org/10.1155/2015/296581

Zapater-Fajari, M., Crespo-Sanmiguel, I., Perez, V., Hidalgo, V., & Salvador, A. (2022). Subjective Memory Complaints in young and older healthy people: Importance of anxiety, positivity, and cortisol indexes. Personality and Individual Differences197, 111768. https://doi.org/10.1016/j.paid.2022.111768

As the author mentioned in the introduction, memory is an important cognitive ability. Thus. Is there any reason not to choose other measures that also focus on participants’ cognition functions as other studies (e.g., Taleb, Ismail, & Abou-Abbas, 2024)?

A: Please, see our previous comment. We added the reference (Taleb et al., 2024) in the introduction and discussion to highlight the importance of other subjective memory tools to analyze convergent validity. Also, we added other references to inform the reader that other validation studies of the EMQ-R were conducted (Ahmadi et al., 2023; Sander et al., 2018; Stančić et al., 2018; Taleb et al. 2024). See Page 2, Lines 63-66; Page 2 and 3, Lines 98-116; and Page 11 Lines 435-440; text highlighted in yellow.

Reference

Ahmadi, A., Hajipour, M., Vojoudi, F., Haresabadi, F., Mashhadi, A., Nahayati, M. A., & Maleki Shahmahmood, T. (2023). Translation, cross-cultural adaptation, and validation of the Persian version of Everyday Memory Ques-tionnaire-Revised (EMQ-R) in patients with multiple sclerosis. Applied Neuropsychology: Adult, 1-9. https://doi.org/10.1080/23279095.2023.2205592

Sander, A. M., Clark, A. N., van Veldhoven, L. M., Hanks, R., Hart, T., Leon Novelo, L., Ngan, E., & Arciniegas, D. B. (2018). Factor analysis of the everyday memory questionnaire in persons with traumatic brain injury. The Clinical Neuropsychologist, 32(3), 495–509. https://doi.org/10.1080/13854046.2017.1368714

Stančić, S., Dimitrijević, S., & Subotić, S. (2018). Evaluation of the Everyday Memory Questionnaire-Revised (EMQ-R). Empirical Studies in Psychology, 9.

Taleb, A., Ismail, A., & Abou-Abbas, L. (2024). Psychometric properties of the Arabic version of the everyday memory questionnaire-revised (EMQ-R) among the Lebanese population. The Clinical Neuropsychologist, 1-18.

Maybe I misunderstood some content of the manuscript, I wonder what is binary (criteria the authors used in the ROC curve analyses for the EMQ-R? Is it the diagnosis for each participant provided by clinical neuropsychologists? If so, I will suggest authors mention it in the manuscript.

A: We appreciate the question and understand the concern. However, we followed the procedure used in other studies (e.g., Teles et al., 2021) and employed a single dichotomous item in which participants reported the presence or absence of memory complaints. We acknowledge that the lack of an assessment by neuropsychologists is a limitation. However, we have made this limitation more explicitly reflected in our discussion (see Page 10, line 423-434, text in highlighted in yellow).

Reference

Teles, S., Ferreira, A., & Paúl, C. (2021). Assessing attitudes towards online psychoeducational interventions: Psychometric properties of a Brief Attitudes Scale. Health & social care in the community29(5), e1–e10. https://doi.org/10.1111/hsc.13227

I also have several concerns about the factor analyses in the manuscript. It looks like the second-order CFA model also has some issues with the correlations between first-level factors. The standardized loadings from the higher factor to the two lower-level factors are both higher than 0.95, which means the correlation between them is still above 0.9. In other words, these two factors are not differentiable in CFA (i.e., similar to the multicollinearity issue the authors mentioned in the first paragraph of section 3.3). Besides, the modification index will only tell researchers the estimated change in global Chisq after releasing one estimate constraint, rather than suggesting any kind of “second-order model”, if the starting model is a typical first-order CFA model. Therefore, I will suggest the authors redo the analyses here and use more precise phrases to describe the procedures of FA. For example, given there is some uncertainty about the proper measurement model for the EMQ-R, I will suggest the authors split the sample into two groups. Using the first group to run EFA and explore the possibilities, and then use data from the second group to verify it with CFA. In addition, the other can include indices that account for model complexity such as AIC and BIC in their report for CFA. These indices are available in Mplusin Mplus if the authors use ML estimators for categorical variables.

A: We appreciate your valuable comments. The entire data analysis was restructured based on the suggestions provided. As recommended, we split the sample into two groups: the first group was used to perform exploratory factor analysis (EFA) and explore the possibilities, while the data from the second group were used to verify the structure with confirmatory factor analysis (CFA). However, we chose to use the WLSMV estimator in our confirmatory analysis, as we considered it to be more robust for ordinal data (e.g., Li et al., 2016). Therefore, with this estimator, it was not possible to determine the AIC and BIC in the CFA report (please see Pages 7-10; text highlighted in yellow). Also, we also made adjustments to the abstract as a result of the changes made in the manuscript (Page 1).

Reference:

Li C. H. (2016). Confirmatory factor analysis with ordinal data: Comparing robust maximum likelihood and diagonally weighted least squares. Behavior research methods48(3), 936–949. https://doi.org/10.3758/s13428-015-0619-7

To benefit the readers, I will suggest the authors directly provide the translated version of EMQ-R in the paper or at least in some online supplementary material (e.g., posted on OSF), rather than having readers further request for it via personal communications.

A: We appreciate your suggestion to make the translated version of the EMQ-R available directly in the article or as supplementary material, instead of relying on personal communications. We agree that this would improve accessibility and convenience for readers. We will consider making it available as Appendix A.

Other minor issues, as the common practices of reporting FA results, I will suggest the authors add means and SDs of 13 items into Table 1. Besides, I will also recommend the authors provide the correlation matrix between these 13 items in the data, so other researchers can replicate the FA results to some degree.

A: Thank you for your insightful suggestions. Regarding your recommendation to include the means and standard deviations (SDs) of the 13 items in Table 2, we would like to clarify that, given the ordinal nature of the items, we initially opted to report the median and interquartile range (IQR) as they provide a more appropriate representation of central tendency and variability for ordinal data (e.g., Cooksey, 2020). However, in response to your suggestion, we have also included the means and SDs in Table 2 to provide additional context. Additionally, the correlation matrix between the 13 items has been included in Appendix B, as suggested, to enable other researchers to replicate the factor analysis results more effectively.

Reference

Cooksey R. W. (2020). Descriptive Statistics for Summarising Data. Illustrating Statistical Procedures: Finding Meaning in Quantitative Data , 61–139. https://doi.org/10.1007/978-981-15-2537-7_5

If all the participants are non-clinical persons, I suggest the authors add this information to the title of the article.

A: Thank you for the suggestion. Now, we added this information in the title of the manuscript (Page 1) and in the introduction (Page 3, Lines 120-121).

Reviewer 2 Report

Comments and Suggestions for Authors

1.      At Line 243 and 251. The term "multicollinearity" might be misused in this CFA situation. The multicollinearity in statistics refers to the high pairwise correlations between independent variables in regression functions. In the CFA model, as shown in Figure 1, the item responses are dependent variables. Line 243 said, "… and no multicollinearity was detected (0.235 ≤ r ≤ 0.703)." It is strange and probably an incorrect sentence. This sentence is addressed with values in Table 2, the descriptive statistics of EMQ-R items. I suppose (0.235 ≤ r ≤ 0.703) describes the range of pairwise correlations between the 13 items (it is also unclear whether it is an inter-item correlation). The inter-item correlation should be nothing related to the term "multicollinearity" because all items are dependent variables in CFA (see Figure 1). Line 251 said, "… a high correlation between latent variables, indicating multicollinearity (r>0.90)." It might also be an incorrect sentence. Although the two latent traits are independent variables, they do not have overlapped dependent variables (i.e., items), so the term "multicollinearity" should not be used here. The high correlation between latent traits is the problem of misspecification of the CFA model, where we might misspecify a multidimensional model for a unidimensional model. Please revise or remove the two incorrect sentences about multicollinearity.  

2.      From Lines 206 to 217. Readers might need more clarification about the true condition when creating the ROC. The ROC is created by calculating the true positive and false positive rates in a contingency table between the predicted and true conditions. The predicted condition is the diagnosis by EMQ-R with hypothesised cut-off scores. What is the true condition in your ROC calculation? I think your true condition would be the "Current memory complaints (yes)" in Table 1. However, such an important variable was not clearly described in the methodology. Please provide a paragraph on methodology to describe how you collected the variable "Current memory complaints (yes)" shown in Table 1. I guess the "perceived memory difficulty" in Line 137 is the "Current memory complaints (yes)" in Table 1. Is my guess correct? Please clarify it. The second problem here is whether "Current memory complaints (yes)" are measured by self-reported questions or by clinic diagnostic results from practitioners. It is better that it was measured by clinic practitioners' evaluation than by self-reported questions because this variable is used as the true condition in ROC analysis. When a participant's self-reported "Do you currently experience memory complaints" can be a true condition of her/his memory complaint condition in ROC, it implies we can use one question to get participants' true conditions (i.e., Do you currently experience memory complaints?) and why we need EMQ-R (13 items) to predict their memory complaints conditions? The ROC analysis setting cut-off scores aims to maximise the true positive rate and minimise the false positive rate of the prediction compared to the true condition. When we can get the true condition by a one-question self-report, why do we need the prediction model by a 13-question self-report? When we get the true condition by clinic practitioners' evaluation, we can claim that our cut-off point is worthy because the diagnosis result from self-report can replace clinic practitioner evaluation, which reduces the cost of diagnosis. In summary, please 1) clarify the variable "Current memory complaints (yes)" in methodology, which is an important variable for ROC calculations, 2) clarify your ROC calculations in methodology, and 3) add a limitation if the variable "Current memory complaints (yes)" is collected by a self-report question.

3.      If possible, could you add a measurement invariance (MI) or differential item functioning (DIF) analysis? The measurement invariance analysis provides construct validation evidence that the psychometrics properties of the measurement construct are consistent between groups. The grouping variables can be gender or "Current memory complaints (yes)" in your study. Please follow the MI analysis guidelines written by Svetina et al. in 2019. I can understand that a small sample size might meet a convergence problem when fitting a multigroup model in MI analysis. I would appreciate it if you could add MI or DIF analysis to this validation work.

Svetina, D., Rutkowski, L., & Rutkowski, D. (2019). Multiple-Group Invariance with Categorical Outcomes Using Updated Guidelines: An Illustration Using Mplus and the lavaan/semTools Packages. Structural Equation Modeling: A Multidisciplinary Journal27(1), 111–130. https://doi.org/10.1080/10705511.2019.1602776

Author Response

Reviewer 2

At Line 243 and 251. The term "multicollinearity" might be misused in this CFA situation. The multicollinearity in statistics refers to the high pairwise correlations between independent variables in regression functions. In the CFA model, as shown in Figure 1, the item responses are dependent variables. Line 243 said, "… and no multicollinearity was detected (0.235 ≤ r ≤ 0.703)." It is strange and probably an incorrect sentence. This sentence is addressed with values in Table 2, the descriptive statistics of EMQ-R items. I suppose (0.235 ≤ r ≤ 0.703) describes the range of pairwise correlations between the 13 items (it is also unclear whether it is an inter-item correlation). The inter-item correlation should be nothing related to the term "multicollinearity" because all items are dependent variables in CFA (see Figure 1).

Our Answer (A): Thank you for your valuable feedback and for pointing out the use of the term "multicollinearity." We appreciate your insightful comments. Upon reviewing the manuscript, we recognize that "multicollinearity" was not the appropriate term in the context of Confirmatory Factor Analysis (CFA). You are correct that multicollinearity refers to correlations between independent variables in regression analysis, while in CFA, we are analyzing item responses, which are dependent variables. We have already revised the manuscript to replace "multicollinearity" with the correct terminology. Specifically, we clarified that the inter-item correlations between the 13 items of the EMQ-R ranged from 0.235 to 0.703, indicating moderate to strong relationships between the items, with no correlation exceeding 0.80. This suggests that the items are sufficiently distinct without significant redundancy (see Page 6, line 277-281; text highlighted in yellow). We believe this revision addresses the concern, and we appreciate your attention to this detail. Thank you once again for your helpful comments.

Line 251 said, "… a high correlation between latent variables, indicating multicollinearity (r>0.90)." It might also be an incorrect sentence. Although the two latent traits are independent variables, they do not have overlapped dependent variables (i.e., items), so the term "multicollinearity" should not be used here. The high correlation between latent traits is the problem of misspecification of the CFA model, where we might misspecify a multidimensional model for a unidimensional model. Please revise or remove the two incorrect sentences about multicollinearity.  

A: Thank you for your very relevant suggestion. Our analysis was revised based on the issues the model raised. This led to a one-factor solution, excluding item 4, based on the comments from another reviewer. Please review the results section with this change (see Pages 7-10; text highlighted in yellow).

From Lines 206 to 217. Readers might need more clarification about the true condition when creating the ROC. The ROC is created by calculating the true positive and false positive rates in a contingency table between the predicted and true conditions. The predicted condition is the diagnosis by EMQ-R with hypothesised cut-off scores. What is the true condition in your ROC calculation? I think your true condition would be the "Current memory complaints (yes)" in Table 1. However, such an important variable was not clearly described in the methodology. Please provide a paragraph on methodology to describe how you collected the variable "Current memory complaints (yes)" shown in Table 1. I guess the "perceived memory difficulty" in Line 137 is the "Current memory complaints (yes)" in Table 1. Is my guess correct? Please clarify it. The second problem here is whether "Current memory complaints (yes)" are measured by self-reported questions or by clinic diagnostic results from practitioners. It is better that it was measured by clinic practitioners' evaluation than by self-reported questions because this variable is used as the true condition in ROC analysis. When a participant's self-reported "Do you currently experience memory complaints" can be a true condition of her/his memory complaint condition in ROC, it implies we can use one question to get participants' true conditions (i.e., Do you currently experience memory complaints?) and why we need EMQ-R (13 items) to predict their memory complaints conditions? The ROC analysis setting cut-off scores aims to maximise the true positive rate and minimise the false positive rate of the prediction compared to the true condition. When we can get the true condition by a one-question self-report, why do we need the prediction model by a 13-question self-report? When we get the true condition by clinic practitioners' evaluation, we can claim that our cut-off point is worthy because the diagnosis result from self-report can replace clinic practitioner evaluation, which reduces the cost of diagnosis. In summary, please 1) clarify the variable "Current memory complaints (yes)" in methodology, which is an important variable for ROC calculations, 2) clarify your ROC calculations in methodology, and 3) add a limitation if the variable "Current memory complaints (yes)" is collected by a self-report question.

A: Thank you for your detailed feedback and insightful comments. Below, we address each of your points:

Clarification of the variable "Current memory complaints (yes)"

In our study, the variable "Current memory complaints (yes)" represents participants' self-reported perception of memory difficulties. It was assessed using a single dichotomous item in which participants were asked: "Do you currently experience memory complaints?" (Yes/No). This item was used to determine the presence or absence of perceived memory difficulties in daily life. We acknowledge that this aspect was not explicitly detailed in the methodology, and this information has now been revised and included in the methodology section (see Page XX, Line Y).

Clarification of ROC calculations

In our ROC analysis, the true condition was defined based on participants' responses to the aforementioned self-reported question on current memory complaints. This procedure has been used in previous studies focused on different research areas (e.g., Teles et al., 2021). However, we recognize the limitations of this approach. Given that this was an online study, we were unable to obtain more precise assessments from a neuropsychologist. Additionally, to our knowledge, there are currently no validated measures in Portugal specifically designed to assess this construct—only tools for evaluating cognitive complaints in general. Therefore, this was the only available alternative.

The predicted condition was determined by the EMQ-R scores using different hypothesized cut-off values. The ROC curve was generated by calculating the true positive and false positive rates from the contingency table, comparing the EMQ-R classification with the self-reported presence of memory complaints.

Acknowledging a limitation

We recognize that using a self-reported question as the true condition in the ROC analysis presents a limitation. Self-reports are inherently subjective and may be influenced by individual perception, emotional state, or cognitive biases. Unlike a clinical diagnosis by healthcare professionals, self-reported memory complaints do not objectively confirm the presence of memory impairment. We have now expanded the discussion of this limitation, emphasizing that while self-reports provide valuable insights, future studies should aim to validate findings with objective clinical assessments.

Reference

Teles, S., Ferreira, A., & Paúl, C. (2021). Assessing attitudes towards online psychoeducational interventions: Psychometric properties of a Brief Attitudes Scale. Health & social care in the community29(5), e1–e10. https://doi.org/10.1111/hsc.13227

If possible, could you add a measurement invariance (MI) or differential item functioning (DIF) analysis? The measurement invariance analysis provides construct validation evidence that the psychometrics properties of the measurement construct are consistent between groups. The grouping variables can be gender or "Current memory complaints (yes)" in your study. Please follow the MI analysis guidelines written by Svetina et al. in 2019. I can understand that a small sample size might meet a convergence problem when fitting a multigroup model in MI analysis. I would appreciate it if you could add MI or DIF analysis to this validation work.

A: We sincerely appreciate the reviewer’s suggestion regarding the inclusion of a measurement invariance (MI) or differential item functioning (DIF) analysis. We fully acknowledge the relevance of such analyses in establishing the consistency of the measurement properties across groups. However, following the recommendation of another reviewer, we revised our analytical approach by splitting the sample to conduct an exploratory factor analysis (EFA) in one subset and a confirmatory factor analysis (CFA) in another. This cross-validation strategy was implemented to enhance the robustness of the factorial structure and to ensure that the model was not overly influenced by a single dataset. Given this methodological change, our primary objective was to confirm the underlying structure of the instrument rather than to examine measurement equivalence across groups at this stage. Additionally, conducting MI or DIF analyses requires sufficiently large subgroup sizes to ensure reliable parameter estimation and meaningful comparisons. Given our current sample distribution, such analyses could introduce convergence issues and limit the interpretability of results. That said, we acknowledge the importance of these analyses and suggest that future studies with larger and more balanced samples could explore the measurement invariance of the scale across different demographic and clinical subgroups.